# The Primary Sclerosing Cholangitis (PSC) Wellbeing Study: Understanding psychological distress in those living with PSC and those who support them

Veronica Ranieri[1,2]*, Eilis Kennedy[1,3], Martine Walmsley[4], Douglas Thorburn[5], Kathy McKay[1,6]

1 Research & Development Unit, Tavistock Centre, Tavistock & Portman NHS Foundation Trust, London, United Kingdom, 2 Department of Clinical, Educational & Health Psychology, University College London, London, United Kingdom, 3 Research Department of Clinical, Educational & Health Psychology, University College London, London, United Kingdom, 4 PSC Support, Didcot, United Kingdom, 5 Sheila Sherlock Liver Centre & UCL Institute of Liver and Digestive Health, Royal Free Hospital, London, United Kingdom, 6 Institute of Psychology, Health and Society, University of Liverpool, Liverpool, United Kingdom

* v.ranieri@ucl.ac.uk

**Data Availability Statement:** Our interview transcripts and coding contain identifiable information such as names and places. We cannot

## Abstract

### Introduction

The impact of living with Primary Sclerosing Cholangitis (PSC) on psychological wellbeing is not well-known. A recent scoping review by the authors found that both depression and anxiety frequently featured in the accounts of those living with the illness. However, less clear were the factors that led to such psychological distress, the impact that the illness had on families and how to best support those living or supporting someone living with the illness. In light of this, the aim of this study was to explore how the illness impacted the lives of both those diagnosed with the illness and those supporting them.

### Method and results

This study adopted a phenomenological approach to understand the subjective experiences of individual participants. A total of 30 individuals took part in Asynchronous Virtual Focus Groups hosted on a Virtual Learning Environment for a four-week period. Chronological narratives of individuals' lived experiences from diagnosis to post-transplant are presented below. These narratives centred upon individuals' and families' experiences of receiving a diagnosis, and adjusting to life post-diagnosis, particularly in regard to their relationships with health professionals and other family members, and in preparing for the possibility of transplant.

### Discussion

The present article provides an in-depth look at how PSC can impact psychological wellbeing, how psychological distress arises and includes advice tailored to individuals, families and health professionals on how to best support each other.

presently anonymise these files due to the covid19 pandemic. Should researchers request access to such data in the future, we will be able to accommodate these once the covid19 pandemic has resolved and we can access office files again. For data access requests, interested researchers may reach out to Dr. Jane Petty, who is the overall research coordinator for the Tavistock's Research & Development team. Her email is j.petty@tavi-port.nhs.uk.

**Funding:** This work was supported by PSC Support grant number 220318VR. The funders did not play a role in the study design, data collection and analysis, decision to publish, or preparation of the manuscript.

**Competing interests:** MW is the Chair of Trustees at PSC Support and an individual living with PSC. DT is a member of the PSC Support Expert Panel. No other authors have competing interests.

# Introduction

Primary Sclerosing Cholangitis (PSC) is a cholestatic illness caused by inflammation and scarring of the liver's bile ducts [1]. Although some individuals may be asymptomatic at first, the illness is associated with burdening symptoms such as itching, fatigue, jaundice and abdominal pain [1, 2]. It is prevalent in 0–16.2 cases per every 100,000 individuals, though prevalence rates appear to be on the increase [1, 3]. At present there is no known cause or medical treatment to halt or slow the progression of the illness, with some of those diagnosed with PSC progressing to end-stage liver disease that requires transplantation or developing cancer [4, 5].

The impact of living with Primary Sclerosing Cholangitis (PSC) on psychological wellbeing is not well-known. A recent scoping review by the authors mapping the available literature on PSC and its plausible relationship with psychological wellbeing, found that individuals living with the illness displayed a greater number of depressive symptoms and poorer wellbeing, often coinciding with stage of liver disease and comorbidity with Inflammatory Bowel Disorder (IBD) [6–10]. Anxiety too was frequent and manifested as fears regarding future malignancy, mortality, and receiving a transplant [7, 11, 12]. Missing from these accounts, however, is an understanding of the causes of such distress, and a chronicle of the experiences and perceptions of those living with PSC and caregivers ranging from diagnosis to life after transplant. For this reason, this study qualitatively examined the lived experiences of both those living with PSC and those informally supporting those living with PSC.

# Materials and methods

## Participants

Participants consisted of individuals aged 18 years or older, who were resident in the UK at the time of participation, and self-identified either as someone living with PSC or a partner, family member or carer of a person living with PSC. Participants were recruited and took part in the study between September-November, 2018. These were purposively sampled according to length of diagnosis, age and whether or not these had experiences of transplantation, in order to ensure that a breadth of viewpoints was included.

A total of 30 individuals took part in the study: 22 were individuals living with PSC, with 8 family members supporting individuals living with PSC. Participants were allocated to one of four focus groups with 6–9 people in each. Three of the focus groups comprised of individuals living with PSC and included 10 males and 12 females. Their ages ranged from 28–74 years, and their lengths of diagnosis at the time of the AVFG ranged from 6 months—20 years. The majority (N = 19) had not received a transplant. Family members supporting those living with PSC were all female: five mothers and three partners of individuals living with PSC. Their ages ranged from 31–60 years and their relative's length of time from diagnosis to the time of the focus group ranged from 6 months—10 years.

## Recruitment

Participants were recruited via PSC Support, a UK-based charity that provides information and support to those affected by PSC, by means of study advertisements placed on its media channels (ie. Facebook, Twitter, PSC Support website). These advertisements featured information relating to the study and its purpose, in addition to the researchers' contact details. All potentially interested participants contacted the researchers using the contact details provided in the advertisements. All those interested and eligible were then invited to take part in an Asynchronous Virtual Focus Group (AVFG). AVFGs are online focus groups where

individuals log onto an online platform to post responses to questions [13, 14]. Participants are typically asked to write one or more responses to each question within a given time limit.

## Setting

AVFGs were hosted on Blackboard, a Virtual Learning Environment (Blackboard) that provided a secure and confidential space for research participants to write and interact with each other [15–17]. This platform was chosen as it allowed the researchers to recruit individuals with a rare illness from across the UK, and participants were able to take part in the study at a time of their choosing and from the comfort of their own home.

## Procedure

Participants were registered onto Blackboard under an alias chosen by them in order to preserve their anonymity. Once registered, they were asked to log-in and familiarise themselves with both the platform and all documents listed as 'essential information' within it. 'Essential information' included: the study's information sheet and separate consent form, contact details for the researchers and support, and netiquette guidelines. Informed consent was necessary for all participants and recorded online as a graded task. Once participants consented, the first focus group question appeared as a discussion topic. Participants were asked to answer a different question each week for four weeks, relating to their experiences of receiving a diagnosis of PSC, contact with health professionals, helpful supports and unhelpful barriers, and advice they would share with someone newly diagnosed. These topics were identified by members of PSC Support, who had lived experience of PSC. For a list of questions, please see Table 1. Participants could post as often as they wished each week, both in response to the question posed and in reaction to other participants' responses. Participants received an email notification for each new question posted. In order to ensure participants' wellbeing during the AVFGs, discussion boards were moderated twice daily by researchers KM and VR.

## Analysis

This study took a phenomenological approach as it was interested in understanding the subjective experiences of individual participants and the meaning that participants attributed to these experiences. Focus group data were downloaded directly as text from Blackboard and coded using NVivo as a data management system [18]. Text scripts were analysed adapting Braun & Clarke's (2006) thematic analysis with features of narrative analysis, as detailed by Riessman (2008) [19, 20]. This involved both sequencing events and experiences, and grouping commonalities and experiences within chronological sequences, and between the two types of

**Table 1. Focus group questions divided according to week.**

| Week | Question(s) |
|---|---|
| Week One | Diagnosing PSC can be difficult. Can you tell us about how you felt when you were first diagnosed? How have your feelings changed between then and now, if at all? Have you experienced an onset of symptoms? What was your experience of these symptoms? |
| Week Two | What was it like dealing with healthcare professionals before your diagnosis? Can you tell us about your experiences dealing with healthcare professionals now that PSC is being treated? Have you been offered emotional support? If yes, what kind of support was offered? |
| Week Three | Since your diagnosis, what's been most helpful for you? What helps you through the tough days? Has there been anything that's been unhelpful? |
| Week Four | What advice would you give someone who has just been diagnosed with PSC? |

participants. Themes and narratives were explored by two researchers (VR and KM), and any disagreements discussed and resolved.

### Ethical approval

This study was approved by the Health Research Authority from London—Queen Square NHS Research Ethics Committee under application number 18/LO/1075.

## Results

The narratives of those living with PSC are presented separately to those supporting someone. These narratives follow a chronological order, beginning with participants' experiences of receiving a diagnosis, living with or supporting someone living with PSC post-diagnosis, to chronicling experiences of transplantation and life post-transplant, where applicable. The themes extrapolated from the data are presented in Table 2.

### Living with PSC

**Experiences of diagnosis.** Diagnosis of PSC began in one of three ways: after abnormal liver function tests were detected in routine blood tests, during routine scans for another serious condition that participants were already living with, or when participants reported increasingly disabling symptoms of unknown origin that led them to seek medical attention. Such help-seeking was sometimes met with attentiveness and concern by a helpful GP who speedily referred the individual for diagnostic testing. Unfortunately, for others, help-seeking was met with dismissal and misdiagnosis whereby symptoms were wrongly attributed to stress, burn out, alcohol misuse, or a less serious physical condition. Such misdiagnosis led to secondary care gatekeeping and prolonged delays in receiving appropriate specialist attention and treatment. This sub-group wrote of being taken more seriously only when symptoms became more advanced, or by changing general practitioner.

**Table 2. Themes emerging from the focus groups divided according to timeline and type of participant.**

| Participant | Timeline | Theme |
|---|---|---|
| *Individuals living with PSC* | *Experiences of diagnosis* | Shock |
| | | Anxiety |
| | | Access to information |
| | | Low mood |
| | | Disclosure |
| | *Change after diagnosis* | |
| | | Support from family, friends and peers |
| | | Contact with health professionals |
| | | Advocacy |
| | | Psychological support |
| | | Preparing for transplant |
| | *Life after transplant* | |
| *Caregivers* | *Reacting to a relative's diagnosis* | |
| | *Adjustment* | |
| | | Psychological impact |
| | | Inclusion by health professionals |
| | | Resilience |

*"The first GP I saw just kept telling me I was young and to stop burning the candle at both ends. That was rubbish. I was struggling to stay awake and going to bed early. He made me feel like I was wasting his time and making my symptoms up."* MC, female, 36

**Shock.**   Diagnosis of PSC often came as a shock. Although some felt cared for and reassured by health professionals, the majority felt that their diagnosis was given to them hastily, sometimes over the telephone, with no comprehensive explanation of the implications of having PSC, other than to inform participants that there was presently no known cause or cure for PSC and that a transplant may be warranted in future. Furthermore, participants recounted having little opportunity to ask further questions or receive adequate signposting for support. Such detachment and lack of guidance exacerbated participants' shock at discovering that they had an incurable and rare illness.

*"I was shocked and confused—I felt so well. I was also amazed that the registrars could drop this bombshell on me, and leave me with no one to talk to and no further date for review— with the next step being a liver ultrasound nearly 5 months later."* WW, female, 34

*"The first specialist I saw had no grasp at all what it is like to be told you have a disease that is chronic and has no cure. He was very blasé and to quote him 'well you're a big fella so I thought you wouldn't have any issues, I am surprised you took it this way'."* BB, male, 45

**Anxiety.**   Such shock soon led to anxiety upon diagnosis. Fears regarding longevity, mortality and the need for or *'threat'* of a future transplant featured heavily in participants' accounts, with diagnosis sometimes compared to a *'death sentence'*. Participants questioned their future, and their ability to remain well and achieve the dreams they aspired to, such as child-bearing or -rearing, particularly when diagnosed young. Concerns about developing or experiencing worsening symptoms or illnesses such as cancer, undergoing further extensive medical procedures, and not being alive to see their children grow older were prevalent across the focus groups.

*"I was newly married and had dreamed of starting a family soon. I felt like my hopes and dreams for the future had been dashed in just a few minutes. I returned home and was inconsolable for some time. I was scared of the thought of a transplant, of being told I may have 5–10 years if I was lucky, but most of all being told it was unlikely I would be able to conceive due to the PSC and couldn't have a family. I was devastated."* MC, female, 36

**Access to information.**   Such anxiety was compounded by a lack of easy access to helpful information and guidance. Participants recognised that many of the medical professionals from whom they received a diagnosis were unfamiliar with PSC, and also unprepared for giving such diagnosis. In the absence of information from medical professionals, participants turned to the internet. The information found on the internet, however, was limited and tended to portray more severe and advanced cases of PSC, thus worsening the worries already experienced by many participants upon diagnosis.

*"You are given this devastating news and then turfed out onto the street—literally- with no offers of support and advice, and only Google to turn to for information—which is always a bad idea in my experience!"* OC, female, 48

**Low mood.**    For many, such fears became intertwined with an onset of depressive symptoms, including physical manifestations such as insomnia, disturbed sleep, and weight loss. Spiralling negative thoughts and feelings of hopelessness and despair from learning that PSC had no cure also came to the fore. Expressions of grief and longing for the life participants felt they had prior to diagnosis appeared frequently and focused on a real or foreseen loss of health, quality of life and independence. In some, such distress was followed by a period of self-blame and a desire to reprioritise life choices or achieve previously held dreams before time ran out.

"*I was 29 and felt like my life had all of a sudden changed direction. I remember taking off my make up one night and thinking "what is the point*?*"* WW, female, 34

"*The first week I was stuck in a spiral of negative thoughts, I slept and slept and slept. It felt like my life was over.*" DW, male, 42

**Disclosure.**    Disclosure was often difficult for it cemented the reality of the diagnosis. Some participants chose not to disclose their diagnosis to anyone other than their partner. Though some reported feeling supported by family members and friends, disclosure sometimes amplified the helplessness participants already felt when members too encountered upsetting descriptions of prognostic outcomes, and experienced similar barriers to accessing trustworthy information. However, such disclosure was sometimes met with short-sightedness, both in appreciating the complexity of the illness and understanding that pronounced symptoms were often not visible.

"*They can't see it—just think you are being lazy or moaning!. . . Tiredness will play a part— which is something people can't see so I think they forget that I have something seriously wrong with me. Don't get me wrong—I'm not looking for sympathy on a daily basis, but sometimes some understanding would be good.*" BG, female, 40

**Change after diagnosis.**    The period following diagnosis was characterised by great change in many. Participants wrote of adjusting their life around how they felt each day, knowing their condition may rapidly deteriorate. Although some remained largely asymptomatic, others experienced symptoms such as fatigue, itching, headaches, brain fog, nausea and abdominal pain. They described being no longer able to '*push through tiredness'* and being unable to remain as physically active due to these symptoms. For many, this led to a process of 'destressing' their lives, resulting in changes to diet and exercise and, for some, ceasing employment. Many, however, also reported a subsequent reduction in social activities leading to isolation. This was particularly felt by those who experienced a loss of consistent employment that provided opportunities for socialisation. Feeling isolated due to the rarity of the illness whilst observing family members and friends working and functioning as 'normal' created feelings of loneliness.

"*Since my diagnosis my fatigue continued. I was forced to cut my hours and end my dream career. I had to change my job. I had become unreliable at work and physically was struggling to drive to work.*" MC, female, 34

*"I am 57 years old and my quality of life now is dreadful. I rarely go out and have become quite isolated as my friends and family all work while I sit at home. The only way I can describe it is I feel that I am actually grieving for the life I had."* RW, female, 57

**Support from family, friends and peers.** Families represented a crucial source of support for many. Participants acknowledged the difficulty family members experienced in providing regular support, particularly for an illness where individuals can be periodically asymptomatic or not show physical signs of being unwell. Although the invisibility of the illness was a difficulty for both, family members were praised for the practical and/or emotional support they provided. During periods where participants experienced incapacitating symptoms, they expressed worry and guilt that they may be holding family members back from taking part in activities they were interested in, and sadness that they were missing out on important family moments or unintentionally neglecting the needs of other family members. Participants, however, felt supported when family members took an interest in PSC and researched dietary plans and treatment, as well as attending medical appointments with them. Family members were cherished when they helped calm participants during moments of anxiety, encouraging them to overcome obstacles and bad days.

Similarly, some participants felt that they could rely on their friends in times of need. Others, however, felt abandoned by friends who either failed to show interest in them and their illness, or misunderstood the impact of PSC on their lives, for example in being unable to maintain scheduled meet-ups.

*"The most stressful aspect has been dealing with people who don't understand what it's like to get a diagnosis of a disease like PSC and how hard it can be at times. Some very longstanding friends just stopped contacting me. This was the toughest thing I've ever had to deal with, including the disease itself and having cancer."* DP, female, 56

In searching for an understanding place to counteract their sense of loneliness, participants turned towards peers who could relate, answer questions and exchange information. Peer groups, such as PSC Support, provided a sense of community where members could talk to each other and feel comforted by observing others living a 'normal' life, especially in the absence of local support groups. However, such groups also had the potential to scare or disturb participants when younger or more advanced cases of PSC were discussed.

*"The Facebook group does have its downside because it makes you very aware of all the 'bad stuff' and you watch people suffering and going through hell, it is particularly disturbing to watch the suffering of the younger members of the group."* EL, female, 68

**Contact with health professionals.** Individuals also approached specialists in order to access medication to alleviate symptoms and receive ongoing care. For some, this was a positive experience, resulting in individuals feeling encouraged and cared for. These participants felt reassured when seeing the same specialist at each appointment who shared their contact details in case of need. They described having a good relationship with their specialist, felt 'lucky' to be treated by them, and 'safe' within their treating team. Specialists were commended when taking a holistic approach to the person's care, particularly when the individual had comorbidities. Those accessing liver teams at regional specialist centres or via private

healthcare, in particular, portrayed their specialist as informed, kind and sympathetic, and felt their illness was monitored regularly, and their queries answered.

> "*Woo! What a change that made to my life. I felt like Wonder Woman compared to what I had been like before. This consultant was extremely engaging, knowledgeable, showed genuine interest in me and how I was.*" JJ, female, 53

> "*The regional centre that I'm being treated at for PSC seems to have PSC aware healthcare professionals and I feel lucky to be treated there. I wouldn't go anywhere else for my care now as it feels like outside of that setting the healthcare professionals are still Googling PSC when I present to them*". FM, male, 32

Most, however, expressed ongoing difficulties in receiving supportive care from a specialist due to a lack of continuity of care within the NHS. Participants often experienced prolonged waiting times between appointments, and met with different doctors at each appointment, whose knowledge and interest in PSC varied widely. Where hepatologists were unavailable, due to the size and geographical location of participants' local hospital, participants were seen and treated by a gastroenterologist for PSC. These, however, tended to be perceived as less knowledgeable and more reluctant to discuss PSC. Doctors were, at times, perceived as in need of training on both PSC itself and on how to sensitively deliver information to those with PSC and their families; adjectives such as 'rude', 'blunt', 'unconcerned', and 'unempathetic' were used. Consequentially, these individuals wrote of needing to 'educate' their doctors or feeling *'abandoned to their fate'*. Whilst one individual felt that they needed to take a break due to over-treatment, many others felt they had to advocate to receive further investigations and support.

> "*My parents, who I had taken with me, had questions that they asked to help me and them, but they were brushed off quickly and not really answered. In fact, my specialist was pretty rude and stand offish. He didn't seem to know much about it either other than the real basics.*" AS, female, 28

Contact with specialists remained infrequent for many, particularly if relatively asymptomatic. Therefore, general practitioners acted as participants' go-to for support between appointments. Similarly to specialists, some GPs were greatly valued by participants, and viewed as open-minded and keen to learn and reach out to specialists when unsure of how to treat. Participants were distressed or apprehensive at the prospect that a supportive GP may retire. Others, however, experienced difficulty accessing a GP or found their GP unhelpful and uninformed. One participant wrote that their GP told them that it was their responsibility patient to 'chase' the consultant.

**Advocacy.** Both disparities in NHS budget allocations across the UK, and poor collaboration and communication across specialist teams and primary care meant advocating to receive appropriate medical attention and support was a common occurrence, both in secondary and primary care. Participants had chased up appointments and test results, prompted physicians for monitoring, and liaised between different hospitals and teams, sometimes across different counties, to ensure that their medical records were up-to-date. Without self-advocacy, appointment scheduling or regular monitoring would fall through, but required energy and an understanding of medical terminology, which participants did not always have.

*"I have been the one chasing up appointments and letters and have often made sure I take my blood results from my GP to my liver consultants or other consultant appts, as I cannot rely on their systems to communicate efficiently. If I didn't do this, then often they would never have seen the information. For me the communication between different trusts and various healthcare professionals is very poor at times."* MC, female, 36

*"The different funding pots of the NHS have a real influence on quality of care–[retracted hospital name] were surprised that my GP was prepared to prescribe one of my anti rejection drugs, as their experience was that many wouldn't. As a patient you shouldn't get caught up in these inter-NHS issues."* OC, female, 48

*"I fear that sometimes it comes down to who's budget is paying for the tests or medication. I know my hospital stopped giving me medication at the hospital and told me to go to my GP, this was so that the cost didn't come from the hospital budget but from the GPs."* JJ, female, 53

**Psychological support.** When asked about whether they received or accessed psychological support via healthcare professionals, most participants revealed that they were often not asked about how PSC impacted them psychologically. Not being asked about their psychological wellbeing, led those living with PSC to question whether physicians fully understood the psychological impact of the illness. Participants felt that particular consultants assumed that the individual was '*handling it*' and, in one instance, reflected sexist beliefs that a diagnosis of PSC would not affect men emotionally. A need for psychological support was picked up in three instances, either by a nurse working on the individual's IBD team who referred them onto a liaison psychiatry team or by GPs, however, never by a member of a liver team.

*"I don't think many consultants—however good they are—understand the psychological impact of having a long term chronic and progressive rare disease, but if they were enabled to understand even a little it would make a huge difference to treatment."* OC, female, 48

In the overall absence of professional psychological support, participants wrote of adopting alternative coping mechanisms that involved resting, distraction, engaging in intellectually stimulating activities and spending time with family and friends. Volunteering helped participants feel that they were making a difference, by assisting others living with PSC and giving back some of the support received from organisations such as PSC Support. Furthermore, becoming more involved in research on PSC, and more informed about PSC empowered participants, reduced their sense of isolation and gave participants hope about the possibility of successful future treatment. Such hope helped participants maintain an optimistic outlook.

**Preparing for transplant.** The prospect of transplantation brought both relief and anxiety to many individuals actively preparing for the procedure. Transplant assessment and 'booking' represented an end to a continuous deterioration in health, as well as an end to waiting for a long-feared surgery. Such anxiety centred upon fears of undergoing the procedure itself, and changes to life, work and wellbeing post-transplant. Transplant was viewed as an insurmountable barrier by those who had to overcome their fear of hospitals and those who associated transplant with death. For some, these fears were partly alleviated by feeling cared for by a knowledgeable specialist who provided clear information and allowed individuals to retain a sense of control over their care.

*"Increasingly in the year leading up to transplant I saw the same person, who I was able to develop a proper relationship with her. This consistency was really important to me—not having to go over my history again and again with someone different."* OC, female, 48

A need for psychological support was also identified by all of those preparing for transplant. Participants spoke of needing someone to help them manage both the 'everyday' and their anxiety and sense of isolation. Such support was again difficult to access. Participants were told that there was no automatic referral pathway and that they must self-refer to mental health services, which required completing questionnaires at a time where both health and energy were at a low point.

"*During the transplant assessment process, I saw. . . a Mental Health Nurse. I was really clear with her that I wanted and needed psychological support, but she freely admitted that there was no route for them to secure that help for me. I did look at the online referral process, but it was so meaningless (with generic questions such as are you suicidal? have you ever wanted to hurt yourself?) that I just couldn't face it. I was feeling so unwell that I just needed it to be a process that required no effort and which could be done for me.*" OC, female, 48

**Life after transplant.** Three participants received a liver transplant at the time of interviewing. Recovery was extensive and arduous as it led to weight loss, weakness and temporary increased dependency on health professionals and family. Nevertheless, after some time, most reported experiencing an improvement in their health and a return to activities previously enjoyed. Yet, this was accompanied by symptom monitoring and a fear that PSC may someday return. At the time of participation, none of the participants re-experienced PSC symptoms post-transplant.

## Supporting those living with PSC

**Reacting to a relative's diagnosis.** Family members tended to express similar reactions to becoming aware of their relative's diagnosis: a grieving process characterised by initial shock about a serious illness, disbelief and denial of the illness itself and its consequences. These also experienced anger and frustration at seeing their relative in pain or imagining their future. Such grief was often compounded by their relative's young age at the time of diagnosis and seeing their young relative lose the aspirations they had prior to illness. Distress and anxiety were a central feature of this experience but were concealed from their relative in an effort to remain a positive and supportive influence. Nevertheless, feelings of sadness, helplessness and powerlessness due to being unable to make their relative feel physically better or decrease their stress, were commonplace.

"*My biggest fear is that this horrible disease will steal my son from me one way or the other.*" QQ, mother, 50

"*I do worry about how he will manage to support himself in the future. He was upset when his brother became a dad this year as he thinks he will never have the energy to look after a family of his own.*" NW, mother, 57

Not all caregivers were present at the time of diagnosis. Similarly to those living with PSC, those who were present at the time that their relative received a diagnosis of PSC recounted feeling incredulous by the lack of information provided by healthcare professionals and the brisk manner with which diagnosis was delivered. They described needing to resort to free search engines to understand what PSC referred to and its long-term outcomes, often with frightening consequences.

*"I. . . had never heard of PSC. The consultant wrote it on a piece of paper, I dropped my son off at university and came home and googled it. I think that was possibly the longest night of my life! I was devastated to read that the survival was 8–10 years and to his day I am astounded that the consultant did not at least attempt to explain to us what PSC is."* NW, mother, 57

**Adjustment.** Diagnosis led to a pronounced period of adjustment in families. Although PSC impacted many families' every-day lives prior to formal diagnosis, there was an ongoing need to accommodate both sudden and routine onsets of symptoms. However, physical symptoms weren't the only concern for caregivers, with many noted substantive psychological symptoms in their relative arising from receiving a diagnosis of PSC. In some circumstances, those living with PSC returned to live in their family homes. When severity of symptoms and care needed exceeded the family's resources, caregivers took steps to temporarily or permanently stop working.

*"I describe it as living with a huge cloud hanging over our heads as symptoms can be so changeable, one week fine and then next week could be hospital admissions!"* CP, wife, 35

*"[Son] seemed to take the news well but over the following months became mentally ill and was diagnosed with depression and anxiety. When we next visited he was so thin that he had to sit down in the shower, very anxious, crying, shaking and struggling to leave the flat. He came home to stay with us and over the next few months built up his eating. . ."* NW, mother, 57

**Psychological impact on caregivers.** Caring for their relative and adjusting to PSC also impacted the psychological wellbeing of family members, with many experiencing symptoms of psychological distress, although few were offered psychological support. In one instance, the caregiver reported being offered antidepressants which they refused and, in another, the caregiver was only able to secure short-term counselling via their workplace. In an effort to counteract such distress, families spoke of shifting their focus to work or children in an effort to distract themselves. Remaining hopeful that a new curative treatment may be found, staying connected with friends and family, and partaking in exercise helped families achieve a positive outlook. However, not all shared their fears in an effort to not burden others.

*"My son's diagnosis totally devastated us. We are both extremely positive people but do now suffer from some anxiety and stress due to worrying about the future. . . I worry all the time when he's ill and I think more negatively than normal. . . even to the point that I'm scared I might lose him. These are when I have really dark days and I don't really share my feelings with anyone, because I don't want to worry family members."* EJ, mother, 50

**Inclusion by health professionals.** Comparably to those living with PSC, families' recollections of their contact with their relative's healthcare professionals varied. Some expressed feeling listened to and included in their relative's care by their relative's treating clinician. However, many other participants highlighted experiences where their concerns were dismissed by health professionals, some of whom cited patient confidentiality as a reason for not keeping caregivers abreast of their relative's illness progression. These caregivers felt the need

to advocate for their relative in order for them to receive adequate care, from professionals who were unfamiliar with PSC.

"*I'm beginning to feel a bit desperate—if I don't chase, my son gets forgotten.*" DF, mother, 52

Families also turned to online support websites led by charities such as PSC Support. Such websites were viewed as an invaluable source of friendly support and up-to-date information on PSC and research into the illness.

"*He is now seen each time by the same senior consultant. He has got to know us really well and is extremely supportive. We have access to contact him anytime. This has made an immense difference to both our emotional well-being.*" EJ, mother, 50

**Resilience.** Although PSC brought a high degree of uncertainty regarding the future, care-givers also spoke of positive changes resulting from their relative's diagnosis, including an increase in resilience and positive outlook. Though these expressed that PSC was '*always at the back of* [their] *mind*', they spoke of accepting the impact of the illness on the family. Support-ing someone with PSC, in some cases, also led families to feel closer to each other in fighting the illness together.

"*I . . . have often thought when I have been under the weather, how on earth does he get up and carry on feeling like that every day!*" NW, mother, 57

"*Our outlook on life has definitely changed we try not to stress about the smaller mundane things that used to seem stressful.*" CP, wife, 35

## Discussion

Diagnosis of PSC often came as a shock to both those diagnosed with it and family members. It was often shared hastily or with no comprehensive explanation by health professionals about what PSC may entail, and with little opportunity to ask further questions or receive ade-quate signposting for support. Furthermore, it was associated with psychological distress in those living with the illness and caregivers, who spoke of ongoing fears regarding longevity and mortality, and about worsening symptoms or further serious illnesses such as cancer. Such distress was compounded by a lack of access to reliable information or specialists and dis-rupted or disjointed care. Both groups spoke of difficulties in adjusting to life post-diagnosis, often due to decreased levels of functioning and health and, in some, a sense of isolation due to reduced opportunities to socialise. In the absence of professional psychological support, partic-ipants turned to other coping mechanisms, such as volunteering and partaking in activities that provided distraction and a sense of intellectual stimulation. Connecting with other mem-bers of their families, friends and peers who understood the illness helped participants remain positive.

### Comparison to previous literature

Although there is dearth of literature that documents the experiences of those living with PSC and their caregivers, many of the findings of this study are comparable to the experiences of those with other rare, serious or progressive illnesses. For instance, the presence of psychologi-cal distress is notable in those living with rare or advanced cancers [21, 22]. Reduced physical

functioning due to disabling symptoms, and reduced wellbeing and functioning in psychological, work and social spheres have been reported in those living with other illnesses [23]. Shock at receiving a diagnosis of an incurable illness and feeling distressed about the progressive nature of it is recognised in the narratives of those living with incurable cancer [24]. Participants' sense of isolation and loneliness is also reflected in the accounts of those living with rare, incurable or painful illness [25–27]. Finally, coping responses such as denial, acceptance, positivity and slowing down the pace of life were reported by individuals who, like many of those living with PSC, had multiple chronic conditions or were living with incurable cancer [24, 28].

## Strengths and weaknesses

This study had four main strengths and four limitations. The use of online focus groups allowed the researchers to access geographically diverse populations within the UK who may have otherwise been unable to participate in the study using conventional qualitative methods. It also allowed participants, who could be very symptomatic, to take part in the study from the comfort of their own home by logging in at a time of their choosing. Such ease of access may have facilitated the researchers in reaching saturation, whereby no new knowledge was emerging. Participants also wrote their own narratives, which was potentially therapeutic in itself and also eliminated the need for transcription. However, the use of this method also reduced participants' spontaneity of speech as these could edit their text prior to submitting it. Participation in the study was literacy and tech-literacy dependent which may have excluded some of our target population. Participants also responded to advertisements in order to participate in the study which may have introduced a self-selection bias in our sample. Furthermore, we posit that saturation was unlikely to have been reached with regard to caregivers' accounts, and narratives regarding transplantation. Participants were not specifically asked about transplantation and fewer caregivers contacted the research team to participate. This information was included in light of the limited available research on these topics, therefore, must be interpreted with caution.

## Implications

The study's findings suggest three key implications. Firstly, from participants' narratives, it is clear that greater cohesiveness and interconnectivity between services is needed to facilitate participants' wellbeing, particularly as many of these presented with other serious comorbidities. Such cohesion should also extend to ensuring continuity of care, whereby participants could be seen by the same specialist who was familiar with their case at each appointment. Secondly, a need for signposting to reliable and accurate information and sources of support around the time of diagnosis, especially for such a rare and 'invisible' illness, would help families familiarise themselves with the illness and feel less isolated. Finally, these narratives spoke to a need for greater access to long-term psychological support, and for psychological wellbeing to be included within the discourse of primary and secondary care appointments.

## Future research and clinical practice

Although this is a significant first step towards understanding the experiences of this group, future research should address a continuing gap in knowledge regarding the experiences of those post-transplant and their caregivers. It should also seek to address how psychological distress could be supported within the constraints of both primary and secondary care by incorporating the views of health and psychological health professionals. Finally, both research and clinical practice could be improved by following the advice in Table 3 below, which was

**Table 3. Advice given by participants targeted at newly diagnosed patients and families, and health professions.**

| For those living with PSC and those supporting them: | ⇒ *Become an expert on PSC*: Research PSC and familiarise yourself with the illness so that you know what symptoms to look out for and when you might need medical or psychological intervention. Share what you learn with those supporting you so that they can better support you. Doing so will likely make you feel more empowered in the face of an uncertain illness progression. |
|---|---|
| | ⇒ *However, be careful when googling PSC*: Some of the information commonly found on the internet may be outdated, inapplicable to you, or may not originate from reputable sources leading to considerable psychological distress. It is better to locate information regarding PSC from trusted resources such as PSC Support, the British Liver Trust, and NHS or equivalent healthcare websites. |
| | ⇒ *Accept that you will likely have bad days*: Participants strongly spoke of needing to accept that the illness may get worse, that there is uncertainty regarding its progression and that it will likely cause limitations because of ill health. Denial of PSC was viewed as a barrier to getting support, especially from peers. Acknowledge that the illness and some of the medications may have a psychological impact on wellbeing. |
| | ⇒ *But don't assume that the worst will happen to you*: PSC affects people in different ways. Some have milder symptoms whilst others may need a transplant. Also, remember that there are many positive stories from people who have received a transplant and live 'normal' lives again. Some of those stories emerged within these focus groups. |
| | ⇒ *Nonetheless, prepare for emergencies*: Easy access to contact details for a GP, consultant or A&E may reduce anxiety regarding what to do in the event of acute illness. |
| | ⇒ *Prepare for appointments*: Formulate and note down a list of questions to ask your treating clinician and bring a notepad to take notes. Also, bring someone supportive with you as you may not recall all of what was said during the appointments, and they can ask questions relating to how best to support you. |
| | ⇒ *Try to build a working relationship with a healthcare professional you can trust*: This can be someone you call in case of ill health, that understands the impact of the illness or is willing to research it, and that you feel comfortable speaking to and asking as many questions as you need. You may see many healthcare professionals inconsistently and may not get on with others, so take a note of the ones you found supportive. Don't assume that these are aware of other possible comorbidities and keep them abreast of your wellbeing. |
| | ⇒ *Advocate for what you need*: Ask for more information, regular monitoring, referral to a specialist and referral to psychological support, if you feel you need these. |
| | ⇒ *Please talk to someone*: Decide who you would feel comfortable disclosing your diagnosis to and try to build a support network. Family and friends who could understand the illness and took the time to listen were often a source of comfort and strength for participants. Peer support from other individuals either living with PSC or supporting others living with PSC was a valuable source of information and encouragement. These expressed feeling less alone when sharing their concerns with peers who could advise and show them that PSC was not the focal point of their lives. Speaking to others, such as family, to let them know what is happening for you may also make you feel more psychologically supported and, in turn, lets them understand the illness better. For some, disclosure to employers and colleagues led to receiving workplace support. Others, however, fear being treated differently in the workplace |
| | ⇒ *Take part in research (if you feel up to it)*: Participants felt that it made them feel like they were part of something larger, where they had a safe space to share their experiences in a way that could help others. |
| | ⇒ *De-stress your life (if you can)*: Focus on spending time doing the things that make you feel happy or well. For some participants, changing jobs or working hours, exercising, taking up hobbies and spending time with family or friends made them feel more psychologically and physically well. |

(*Continued*)

**Table 3.** (Continued)

| Advice for healthcare professionals | ⇨ *Show sensitivity towards patients and families at the time of diagnosis*: Many patients and families will be in shock at the news. Some patients will be asymptomatic at the time of diagnosis and will not have suspected that they were unwell. Many will likely be distressed by the rarity and precarity of the illness. Please provide patients and families with an empathic ear and an opportunity to ask questions. |
|---|---|
| | ⇨ *Get informed if you aren't familiar with PSC*: Participants acknowledged that PSC is a rare illness and that most healthcare professionals will be unfamiliar with PSC. However, these also found that informed professionals were a notable source of support. Don't assume that someone with high liver function test values is misusing alcohol. |
| | ⇨ *Signpost patients and their families to sources of information and support*: All patients and families expressed a need for more information at the time of diagnosis. In the absence of such information from their doctors, they often resorted to using search engines which may show false and/or damaging information. A need for information may also arise between appointments. Familiarise yourself and have a leaflet ready for patients and families to take home with them. Also, signpost patients and families to appropriate websites that focus on liver health. |
| | ⇨ *Speak to patients and families about the psychological impact of PSC*: PSC, like other long-term illnesses, may affect individuals' psychological wellbeing. Symptoms of depression and anxiety were commonly spoken about in participants' narratives. However, they were rarely spoken about or treated adequately by health professionals, according to patients and families. |

formulated by participants, and aimed at improving the psychological wellbeing of other individuals with PSC, caregivers and healthcare professionals.

## Acknowledgments

The authors would like to thank all the participants who shared their experiences with us.

## Author Contributions

**Conceptualization:** Eilis Kennedy, Martine Walmsley, Douglas Thorburn, Kathy McKay.

**Data curation:** Veronica Ranieri.

**Formal analysis:** Veronica Ranieri.

**Funding acquisition:** Eilis Kennedy, Martine Walmsley, Douglas Thorburn.

**Investigation:** Veronica Ranieri.

**Methodology:** Veronica Ranieri, Kathy McKay.

**Project administration:** Veronica Ranieri.

**Supervision:** Eilis Kennedy, Kathy McKay.

**Writing – original draft:** Veronica Ranieri.

**Writing – review & editing:** Veronica Ranieri, Eilis Kennedy, Martine Walmsley, Douglas Thorburn, Kathy McKay.

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
