## [Decision Letter · Decision Letter 0]

3 Feb 2020

PONE-D-19-23248

The Primary Sclerosing Cholangitis (PSC) Wellbeing Study: Understanding psychological distress in those living with PSC and those who support them

PLOS ONE

Dear Dr Ranieri,

Thank you for submitting your manuscript to PLOS ONE. After careful consideration, we feel that it has merit but does not fully meet PLOS ONE’s publication criteria as it currently stands. Therefore, we invite you to submit a revised version of the manuscript that addresses the points raised during the review process.

We would appreciate receiving your revised manuscript by Mar 19 2020 11:59PM. To enhance the reproducibility of your results, we recommend that if applicable you deposit your laboratory protocols in protocols.io, where a protocol can be assigned its own identifier (DOI) such that it can be cited independently in the future. For instructions see: http://journals.plos.org/plosone/s/submission-guidelines#loc-laboratory-protocols

We look forward to receiving your revised manuscript.

Kind regards,

Janhavi Ajit Vaingankar

Academic Editor

PLOS ONE

Journal Requirements:

2. We note you have included a table to which you do not refer in the text of your manuscript. Please ensure that you refer to Table 1 in your text; if accepted, production will need this reference to link the reader to the Table.

Additional Editor Comments (if provided):

The wider readership of the journal will benefit from further information on PSC in the introduction. Some context on how this condition could be unique or of significance in comparison to other progressive conditions and particularly in relation to psychological distress should be added.The wider readership of the journal will benefit from further information on PSC in the introduction. Some context on how this condition could be unique or of significance in comparison to other progressive conditions and particularly in relation to psychological distress should be added.

Reviewers' comments:

Reviewer's Responses to Questions

**Comments to the Author**

1. Is the manuscript technically sound, and do the data support the conclusions?

Reviewer #1: Yes

Reviewer #2: Yes

2. Has the statistical analysis been performed appropriately and rigorously? 

Reviewer #1: N/A

Reviewer #2: N/A

3. Have the authors made all data underlying the findings in their manuscript fully available?

Reviewer #1: No

Reviewer #2: Yes

4. Is the manuscript presented in an intelligible fashion and written in standard English?

Reviewer #1: Yes

Reviewer #2: Yes

5. Review Comments to the Author

Reviewer #1: This is a very interesting study examining the experiences of PSC patients and their relatives using qualitative research methods. This is a quite understudied patient group at least from a mental health and health care point of view. As a mental health professional working for over 15 years in a hospital setting and especially in the field of transplantation and gastroenterology and dealing often with patients with PSC I found the insights of the paper very useful and reflecting the clinician experience. The study design and data collection method are creative and make use of technology to overcome obstacles which otherwise make it difficult to study such a population. In general terms the paper is written clearly and it presents a cohesive narrative that can be helpful to clinicians and health care services professionals or even policy makers. Even though I believe this is a study that should be published, there are several minor or majors issues that in my opinion need to be addressed by the authors before the paper is ready for publication.

Regarding the “introduction”:

1. It would round up the introduction and would be very useful to the readers not very familiar with PSC to describe what it exactly is, the course of the disease and symptoms and its prevalence. Otherwise the scope of the study is well described in the introduction.

Regarding the “Methods”

2. Please add a table describing the sample; sociodemographic data and characteristics related to the disease. I guess you need to do this separately for patients and for relatives.

3. Please describe the methods of data collection Asynchronous Virtual Focus Groups and give a reference if possible

4. The “interview guideline”/questions used by the researchers for the Focus groups are missing. Please add a Table or paragraph with this information

5. Please explain how you developed the questionnaire/questions.

6. It would be important to know the extent/size of the material/text that was analysed. How many pages? Words? Also where there any conflicts in communication or interaction? Did the researchers/moderators have to intervene? And if yes, why and how? And how could this have affected the data?

7. Regarding the data analysis you mention “thematic narrative analysis” and even though you describe it, it remains a bit vague and there is no reference regarding this method of analysis. Please add a reference.

Regarding “Results”

8. You present the sample demographics in the results.. I would suggest putting part this in methods.

9. The results are clearly written, there is a cohesive narrative, it is captivating to read and the text passages add value to the narrative. Nevertheless, I feel the way you have presented the results in terms of “chronological” order (“living with PSC, diagnosis, life after diagnosis, preparing of transplantation, life after transplant) helps the overall narrative by creating an overall thread, but does not do justice to very insightful findings and the richness of the results, that seem to go under in the general chronological categories. A solution to this would be to either add the subcategories of your analysis in the text or highlight (bold or cursive) the subcategories (e.g. grief, lack of support, problematic communication with physicians) within the text. In this way the content of the findings would become more clear and could function as a take home message to the reader.

10. The section "life after transplant" is very thin compared to the rest of the paper.

11. Table 1 is interesting but it appears in the text out of the blue. You need to embed it and explain this better

Regarding the “Discussion”

12. The discussion is very short and weak and it needs to be elaborated. It is basically just a paragraph summarising the findings arguing they cannot be discussed as there have been no previous similar studies to compare. I beg to differ regarding this. I think basically almost each of the subcategories of the results/analysis can be discussed e.g. in relation to other literature (if not studies regarding PSC, it can be studies of similar patient groups); or in relation to weaknesses of the health care system; the attitude of professionals; the role of the support system; the relatives support etc.

13. You can remove the heading “summary of findings from the discussion”; It does not add anything

I checked "no" in question 3 for reviewers regarding the availability of the data, because the authors mentioned they are in the manuscript, but I could unfortunately not see any complete data file attached to the manuscript.

Reviewer #2: This study explores lived experiences of people with PSC in relation to their psychological distress. The manuscript is well written, methodological approach is novel and the results section pertaining to patients’ experiences is highly informative. I offer further suggestions for improvements in the content for consideration.

1. It would be advisable to state the epistemological approach in the abstract and methods section of the manuscript clearly, whether it was interpretative or phenomenological. Based on the framework mapping and style of presentation it comes across as phenomenological. However, it needs to be specified and justified.

2. Introduction: Given the rarity of this condition, it may be useful to add some epidemiological information on PSC. For example, how common it is (prevalence) in the UK or globally and why it may be important to identify sources of distress in this population.

3. Methodology: It is generally well described. But I had several unanswered questions as I read through the content

- Interview guide / weekly questions asked to the participants should be included in the manuscript

- Participants – what stage of the condition were they in? It will be useful to add this to understand some context to the discourse. It is also mentioned that most were picked up during scans for another serious condition. How was it ensured that the content related only to PSC and not for other similar/ more serious conditions?

- Only 3 participants had experience with transplant. How was this discussion conducted in a group with the rest? Was there adequate saturation in the content?

- Authors should comment on the overall assessment of data or thematic saturation in the manuscript.

- I am not sure about the inclusion of caregivers in this work – only 8 cares were included. Did they participate in the same forum/discussion platform? I do not believe this number would be adequate to reach saturation. Were the caregivers relatives of the patient-participants and if so, were there any duplicity issues in the discussions? how were these assessed and addressed? Given the length of the manuscript and the lack of saturation, I would like the authors to consider focusing only on patients. Or compressing the content from caregivers in very brief 1-2 paragraphs (highlighting the lack of saturation as a major limitation in the discussion)

- Elaborate the role of the moderators during the discussions. How was adequate participation from all discussants ensured? Did they participate in the discussions?

4. Results: Although well described, results section is rather long. It may be beneficial to provide a quick summary or schematic representation of the themes and subthemes. Also useful to add the age of the participants attributed to the verbatim quotes.

Some content needs rephrasing for further clarification:

- “However, such groups also had the potential to scare or disturb participants when younger or more advanced cases of PSC were discussed” - provide a quote to explain it

- “Both disparities in NHS budget allocations across the UK, and poor collaboration and communication across specialist teams and primary care meant advocating to receive appropriate medical attention and support was a common occurrence, both in secondary and primary care.” - ‘disparities in NHS budget allocations across the UK’ – were these discussed by the participants? Good to provide 1-2 quote to explain it.

- “These questioned whether physicians understood the psychological implications of living with PSC.”- this is not clear. Please check.

- “The first specialist I saw had no grasp at all what it is like to be told you have a disease that is chronic and has no cure. He was very blasé and to quote him ‘well you're a big fella so I thought you wouldn't have any issues, I am surprised you took it this way’.” BB, male – Should this be part of the ‘experience of diagnosis’ theme?

In addition, as stated previously, please re-assess or justify the thematic analysis around transplantation. It is not clear to whom these experiences relates to.

Likewise, under-saturation of ‘Support’ theme needs to be carefully considered. Possibly significantly reduce the content and remove the theme on ‘support through transplantation’ (only 1 carer experience)

Indicate the reference to Table 1.

5. Discussion: Needs further depth and thought. Much of the discussion is a repetition of the results. Although literature in PSC could be limited, authors should draw some parallels with other severe conditions. I gather psychological distress is universal in all severe health conditions, be it cancer, dementia or other GI conditions. It is important to relate it to the study aims and highlight the relevance and contrition of this work in the purview of PSC care and management. Authors provide some recommendations, whoever these could be possibly strengthened by drawing upon results from other studies in similar conditions.

Minor corrections:

Correct the second sentence in the abstract. It should be “a recent scoping review by the authors..”

Last sentence in the introduction: Change to past tense.

“One participant wrote that their GP told them that it was the patient’s responsibility patient to ‘chase’ the consultant” – delete patient after responsibility

6. PLOS authors have the option to publish the peer review history of their article (what does this mean?). If published, this will include your full peer review and any attached files.

Reviewer #1: No

Reviewer #2: No

---

## [Author Response · Author response to Decision Letter 0]

21 Apr 2020

Please see attached response to reviewers

---

## [Decision Letter · Decision Letter 1]

20 May 2020

PONE-D-19-23248R1

The Primary Sclerosing Cholangitis (PSC) Wellbeing Study: Understanding psychological distress in those living with PSC and those who support them

PLOS ONE

Dear Dr Ranieri,

Thank you for submitting your manuscript to PLOS ONE. After careful consideration, we feel that it has merit but does not fully meet PLOS ONE’s publication criteria as it currently stands. Therefore, we invite you to submit a revised version of the manuscript that addresses the points raised during the review process.

We would appreciate receiving your revised manuscript by Jul 04 2020 11:59PM. To enhance the reproducibility of your results, we recommend that if applicable you deposit your laboratory protocols in protocols.io, where a protocol can be assigned its own identifier (DOI) such that it can be cited independently in the future. For instructions see: http://journals.plos.org/plosone/s/submission-guidelines#loc-laboratory-protocols

We look forward to receiving your revised manuscript.

Kind regards,

Janhavi Ajit Vaingankar

Academic Editor

PLOS ONE

Reviewers' comments:

Reviewer's Responses to Questions

**Comments to the Author**

1. If the authors have adequately addressed your comments raised in a previous round of review and you feel that this manuscript is now acceptable for publication, you may indicate that here to bypass the “Comments to the Author” section, enter your conflict of interest statement in the “Confidential to Editor” section, and submit your "Accept" recommendation.

Reviewer #2: All comments have been addressed

2. Is the manuscript technically sound, and do the data support the conclusions?

Reviewer #2: Yes

3. Has the statistical analysis been performed appropriately and rigorously? 

Reviewer #2: N/A

4. Have the authors made all data underlying the findings in their manuscript fully available?

Reviewer #2: Yes

5. Is the manuscript presented in an intelligible fashion and written in standard English?

Reviewer #2: Yes

6. Review Comments to the Author

Reviewer #2: Thanks for addressing the comments and suggestions carefully. Research in this area is rare and its useful to understand PSC sufferers' experiences. I have one additional suggestion - authors could look at the limitations section carefully and elaborate it a bit further based on some of my and other reviewer's comments that couldn't be addressed due to lack of data. Rest of the manuscript reads well.

7. PLOS authors have the option to publish the peer review history of their article (what does this mean?). If published, this will include your full peer review and any attached files.

Reviewer #2: No

---

## [Author Response · Author response to Decision Letter 1]

28 May 2020

We thank the reviewer for this comment. We have expanded and added to our limitations as follows: 

“Strengths and weaknesses

This study had four main strengths and four limitations. The use of online focus groups allowed the researchers to access geographically diverse populations within the UK who may have otherwise been unable to participate in the study using conventional qualitative methods. It also allowed participants, who could be very symptomatic, to take part in the study from the comfort of their own home by logging in at a time of their choosing. Such ease of access may have facilitated the researchers in reaching saturation, whereby no new knowledge was emerging. Participants also wrote their own narratives, which was potentially therapeutic in itself and also eliminated the need for transcription. However, the use of this method also reduced participants’ spontaneity of speech as these could edit their text prior to submitting it. Participation in the study was literacy and tech-literacy dependent which may have excluded some of our target population. Participants also responded to advertisements in order to participate in the study which may have introduced a self-selection bias in our sample. Furthermore, we posit that saturation was unlikely to have been reached with regard to caregivers’ accounts, and narratives regarding transplantation. Participants were not specifically asked about transplantation and fewer caregivers contacted the research team to participate. This information was included in light of the limited available research on these topics, therefore, must be interpreted with caution.”

---

## [Editor Report · Decision Letter 2]

1 Jun 2020

The Primary Sclerosing Cholangitis (PSC) Wellbeing Study: Understanding psychological distress in those living with PSC and those who support them

PONE-D-19-23248R2

Dear Dr. Ranieri,

We are pleased to inform you that your manuscript has been judged scientifically suitable for publication and will be formally accepted for publication once it complies with all outstanding technical requirements.

With kind regards,

Janhavi Ajit Vaingankar

Academic Editor

PLOS ONE

---

## [Editor Report · Acceptance letter]

25 Jun 2020

PONE-D-19-23248R2 

The Primary Sclerosing Cholangitis (PSC) Wellbeing Study: Understanding psychological distress in those living with PSC and those who support them 

Dear Dr. Ranieri:

I'm pleased to inform you that your manuscript has been deemed suitable for publication in PLOS ONE. Congratulations! Your manuscript is now with our production department. 

Kind regards, 

on behalf of

Ms Janhavi Ajit Vaingankar 

Academic Editor

PLOS ONE